# Spontaneous Bacterial Peritonitis among Cirrhotic Patients: Prevalence, Clinical Characteristics, and Outcomes

**DOI:** 10.3390/jcm11010227

**Published:** 2021-12-31

**Authors:** Naim Abu-Freha, Tal Michael, Liat Poupko, Asia Estis-Deaton, Muhammad Aasla, Omar Abu-Freha, Ohad Etzion, Lior Nesher

**Affiliations:** 1The Institute of Gastroenterology and Hepatology, Soroka University Medical Center, Beer-Sheva 84105, Israel; ohadet34@yahoo.com; 2The Faculty of Health Sciences, Ben-Gurion University of the Negev, Beer-Sheva 84105, Israel; nesherke@bgu.ac.il; 3Department of Public Health, The Faculty of Health Sciences, Ben-Gurion University of the Negev, Beer-Sheva 84105, Israel; tal.michael0@gmail.com; 4Medical School for International Health, Ben-Gurion University, Beer-Sheva 84105, Israel; liat.poupko@gmail.com; 5Division of Internal Medicine, Soroka University Medical Center, Beer-Sheva 84105, Israel; asia.estis@gmail.com (A.E.-D.); aslamohammad491@gmail.com (M.A.); omar_20099@hotmail.com (O.A.-F.); 6Infectious Disease Institute, Soroka University Medical Center, Beer-Sheva 84105, Israel

**Keywords:** cirrhosis, SBP, multidrug resistance, mortality

## Abstract

(1) Background: Spontaneous bacterial peritonitis (SBP) is a feared complication of liver cirrhosis. We investigated the prevalence of SBP, positive ascitic fluid cultures, and risk factors for mortality. (2) Methods: A retrospective analysis of all patients with cirrhosis hospitalized or in follow-up in a single center between 1996 and 2020. The clinical data, long-term complications, and mortality of SBP patients were compared with those of non-SBP patients. Ascitic fluid positive culture was compared with those without growth. (3) Results: We included 1035 cirrhotic patients, of which 173 (16.7%) developed SBP. Ascitic fluid culture growth was found in 47.4% of the SBP cases, with Escherichia coli bacteria detected in 38%, 24.4% grew ESBL-producing bacteria, and 14.5% displayed multidrug resistance. In a Cox regression model, SBP, male sex, prolonged INR at diagnosis, and hepatocellular carcinoma were found to be risk factors for mortality in cirrhotic patients. The long-term all-cause mortality was 60% in non-SBP and 90% in SBP patients. (4) Conclusions: Only a minority of cirrhotic patients developed SBP, 47.4% of which had positive ascitic fluid cultures with high antibiotic resistance. Growth of ESBL and multidrug resistant organisms is becoming more frequent in the clinical setting, reaching SBP mortality of 90%.

## 1. Introduction

Infection is one of the most important causes of acute decompensation and death in liver cirrhosis (LC) [1]. It is well established that 30% to 50% of cirrhotic patients either have preexisting bacterial infections when they were hospitalized or acquired them during this period. Such cases were responsible for up to 25% of deaths in this patient population [2,3,4,5,6,7,8]. Increasing bacterial resistance in general infections (e.g., pneumoniae) and cirrhosis-specific infections (i.e., spontaneous bacterial peritonitis) were reported [2,3,4,5,6,7,8].

A number of general risk factors (e.g., previous hospitalization and nosocomial infection) and specific risk factors (e.g., use of quinolones as SBP prophylaxis) for resistant infections have been identified in cirrhosis patients [2,3]. There has been a shift in favor of gram-positive infections, especially in SBP, which has been attributed to invasive procedures performed during hospitalization [7,8,9].

Ascitic fluid infections can be difficult for infectious disease specialists and clinicians to characterize and treat. There are several known risk factors for SBP, including upper gastrointestinal bleeding, low ascitic protein concentration (<1.5 g/dL), and a history of prior episodes, however, ascitic fluid cultures are frequently negative in patients with suspected SBP [10].

In this study, we aimed to investigate the prevalence of SBP, clinical characteristics, long-term outcomes, and risk factors for ascitic fluid culture growth, and the evolution of antimicrobial resistance profiles throughout the last 24 years. 

## 2. Materials and Methods

For this retrospective study, demographic and clinical data were retrieved from computerized databases of internal medicine departments and the clinic of gastroenterology and hepatology at Soroka University Medical Center (SUMC) between the years 1996 and 2020. Data included etiologies of cirrhosis, laboratory values, complications, and death. The Child-Pugh score for severity of cirrhosis in each patient was calculated retrospectively for time of diagnosis, and again at the time of data collection or time of death if the patient already expired at the time of data collection. Patients were diagnosed as suffering from cirrhosis based on liver biopsy, fibroscan, fibrotest, or the presence of typical clinical findings of cirrhosis, such as ascites or esophageal varices.

SBP was defined by the presence of a polymorphonuclear (PMN) cell count over 250/mm^3^ in the aspirated ascites fluid. Microbiological data was collected and included resistance patterns to antibiotics according to the CLSI classification (CLSI, performance standards for antimicrobial susceptibility testing, CLSI supplement M100), multidrug-resistant bacteria were defined as non-susceptible to at least one antimicrobial agent in three or more drug classes, and extended spectrum beta lactamase (ESBL) was defined as resistant to 3rd generation cephalosporins [11]. We excluded pediatric and noncirrhotic patients, and those that had experienced an episode of secondary peritonitis (defined clinically or by polymicrobial cultures). We assessed several other variables, including previous admission or antibiotic treatment within the last six months, hepatorenal syndrome, and intensive care unit (ICU) admission.

### Statistical Analysis

We described patient characteristics as mean ± SD for continuous variables and as percentages for categorical variables. To compare groups, we used the chi-squared test for categorical variables and Student’s *t* test for continuous variables. Continuous variables that were not normally distributed were compared in the Kruskal Wallis test. *p*-values less than 0.05 were considered statistically significant, and variables found to be significantly associated with mortality were gradually added to a COX regression model, where hazard ratio (HR) was calculated for each. Statistical analysis was performed using ‘R statistics’ (R Core Team, 2019) with the packages ‘lubridate’, ‘dyplr’, ‘survival’, and ‘ggplot2’. The study was carried out in accordance with the principles of the Helsinki Declaration, approval and a waiver of informed consent was granted by the Institutional Review Board (0285-18-SOR). 

## 3. Results 

### 3.1. Patients

Our cohort of cirrhotic patients included 1035 patients, of which 173 (16.7%) were diagnosed with SBP according to neutrophil count in ascitic fluid. Of those, only 82 (47.4%) had positive ascitic fluid cultures. The demographic and clinical characteristics of our study populations are summarized in Table 1.

No significant differences were found regarding gender, age, age at diagnosis, ethnicity, or cause of cirrhosis, between the two groups. However, esophageal varices and esophageal variceal bleeding were more common among SBP patients compared with non-SBP patients; 71% vs. 53% and 41% vs. 26% (*p* < 0.001), respectively. 

Lower hemoglobin, platelets, albumin, and higher bilirubin and INR were found among the SBP group (*p* < 0.001). Significantly higher all-cause mortality was found among SBP patients compared with non-SBP patients; 91% vs. 60% (*p* < 0.001). 

### 3.2. Positive vs. Negative Ascitic Fluid Culture 

The data of patients with positive ascitic fluid cultures compared with those who did not grow any bacteria in cultures is presented in Table 2. Among patients with positive cultures, there was greater representation of males vs. females (72% vs. 55%; *p* = 0.018), younger age at diagnosis (58 years vs. 62 years; *p* = 0.033), and esophageal varices bleeding (49% vs. 29.7%; *p* = 0.034). No significant differences between groups were found regarding etiology of cirrhosis, HCC, or death. Laboratory values are summarized in Table 2. Significantly lower hemoglobin, decreased platelets, and prolonged INR were found among cirrhotic patients with positive ascitic cultures. Among the risk factors for positive culture, there was no significant difference between the groups regarding ICU admission, surgery six months prior to diagnosis, ischemic heart disease, malignancy, cerebrovascular accident, hospitalizations, or antibiotic therapy six months prior to diagnosis. 

### 3.3. Microbial Growth in Ascitic Fluid Culture 

Table 3 summarizes bacterial growth and antimicrobial resistance patterns. The most common organism grown was *E. coli*, found in 31 (37.8%) patients, followed by coagulase-negative staphylococci in 13 (16%) patients. Other bacteria were less common (Table 3). Among the patients with positive ascitic fluid cultures, ESBL-producing bacteria were identified in 20 (24.4%) patients. The most common antibiotic resistance in our cohort was found in *E. coli*, of which 52% were ESBL-producing and 23% resistant to gentamicin. Detailed data are shown in Table 3. Multidrug resistant strains were found among 12 (14.5%) patients, including ESBL-producing bacteria and two other antibiotics classes. 

### 3.4. Survival and Risk Factors for Mortality

Cirrhotic patients with SBP had significantly lower long-term survival rates compared to those without SBP, as depicted in Figure 1. The COX regression model for mortality of cirrhotic patients is shown in Table 4. SBP, male sex, prolonged INR at diagnosis, albumin lower than 3.5 gr/dL, anemia, and HCC are significant risk factors for death among patients with cirrhosis

## 4. Discussion

There were several central findings in our cohort study. First, our study confirms that only a small portion (16.7%) of cirrhotic patients develop SBP. Second, ascitic fluid culture growth was found in only 47.4% of patients with SBP. Third, as in previous studies, *E. coli* was the most common organism found in ascitic fluid, displaying resistance to several antibiotics. Fourth, we found that SBP, male sex, prolonged INR at diagnosis, and HCC were significant risk factors for mortality of cirrhotic patients. Mortality was found in up to 90% of SBP patients, compared with 60% of non-SBP cirrhotic patients.

In our cohort of 1035 cirrhotic patients, only 173 (16.7%) were diagnosed with SBP, and 81 (47%) had acute renal failure as an acute complication of the SBP. There were 82 (47.4%) SBP patients that had positive ascitic cultures and 91 (52.6%) had culture-negative neutrocytic ascites. In a prospective multinational study including 46 centers (1302 patients), the prevalence of SBP was 27% among hospitalized cirrhotic patients, and 39% had positive cultures [12].

In the present study, we investigated several risk factors which could affect the frequency of bacterial growth in the ascitic fluid, including hospitalization, ICU admission, and antibiotic therapy or surgery within six months prior to the SBP episode; all these were not found to be risk factors for bacterial growth. However, the relatively small size of our cohort of 82 patients with bacterial growth makes it difficult to draw conclusions regarding these risk factors. 

Consistent with other studies [13,14], where gram-negative bacteria dominate the composition of flora in ascitic fluid (*E. coli*, in particular), our study showed growth of *E. coli* in about 38% of positive ascitic fluid cultures, 52% (16 of 31) being ESBL-positive and 42% (13 of 31) displaying quinolone resistance. ESBL resistance is of particular interest, since third generation cephalosporins are recommended as first-line empirical treatment for SBP, per international guidelines [15,16]. In the present study, we observed an alarmingly high rate of positive cultures harboring resistant organisms to the recommended empirical therapy; these results are consistent with data from the aforementioned international study [12]. In contrast, reports of resistance to quinolones are variable throughout the literature [17]. In our study, 27% of positive cultures showed resistance to quinolones; 20 (24.4%) of the 82 patients with positive cultures had ESBL-producing bacteria, most of which (75%) grew in the last decade. The increasing prevalence of ESBL-producing bacteria and quinolone resistance, variability of culture growth in ascitic fluid (only 47.4% of SBP patients in our cohort), and inadequate data on susceptibilities pose a complex challenge for clinicians in choosing empirical and targeted therapies. With the global emergence of multidrug resistance reaching 34% in some studies [12,14], awareness of local antimicrobial susceptibilities is key to the successful treatment of SBP infections. 

With respect to mortality, individuals with advanced and decompensated liver cirrhosis with SBP are an extremely vulnerable population. Upon long-term follow-up, we found that mortality of cirrhotic patients with SBP was significantly higher (90%) in comparison to non-SBP patients (60%). Previous studies investigating short-term mortality of cirrhotic patients (with follow-up of one to six months) have reported mortality ranging between 24–49% [14]. The explanation of a high rate of mortality in our study, in comparison the previous studied, could be contributed to the increasing multidrug resistance strains observed in the last years. Moreover, in our study, we collected data for long term follow up, longer than previous studies, and reaching more than 20 years in part of the patients. 

SBP is the most common bacterial infection and one of the feared complications in patients with ascites, which is associated with increased mortality [17]. In our study, SBP, male sex, prolonged INR at diagnosis, and diagnosis of HCC are all independent risk factors for mortality among cirrhotic patients. The high general mortality of cirrhotic patients, supported by our data and the literature, delineates the importance of cirrhosis prevention. Many causes of cirrhosis, such as hepatitis C, non-alcoholic fatty liver disease, and alcoholic liver disease, are treatable or preventable in the era of direct-acting antiviral drugs and ample resources for patient education about lifestyle modifications.

The strengths of our study are the large size of our cohort of cirrhotic patients in a single center and the long-term follow-up of complications and mortality. However, this study also has several limitations, including its retrospective design, lack of data regarding SBP treatment in the course of hospitalization, and that the data included all-cause mortality and not liver-related mortality. 

## 5. Conclusions

A minority of cirrhotic patients in our cohort developed SBP, of which only 47% had positive ascitic fluid cultures with relatively high antimicrobial resistance. Policy for empirical treatment of SBP should be addressed and reevaluated regularly on a local level, especially when a high rate of ESBL producing pathogens is observed. SBP, male sex, prolonged INR at diagnosis, and HCC are all significant risk factors for mortality in cirrhotic patients, with mortality up to 90% among SBP patients, compared with 60% among non-SBP patients.

## Figures and Tables

**Figure 1 jcm-11-00227-f001:**
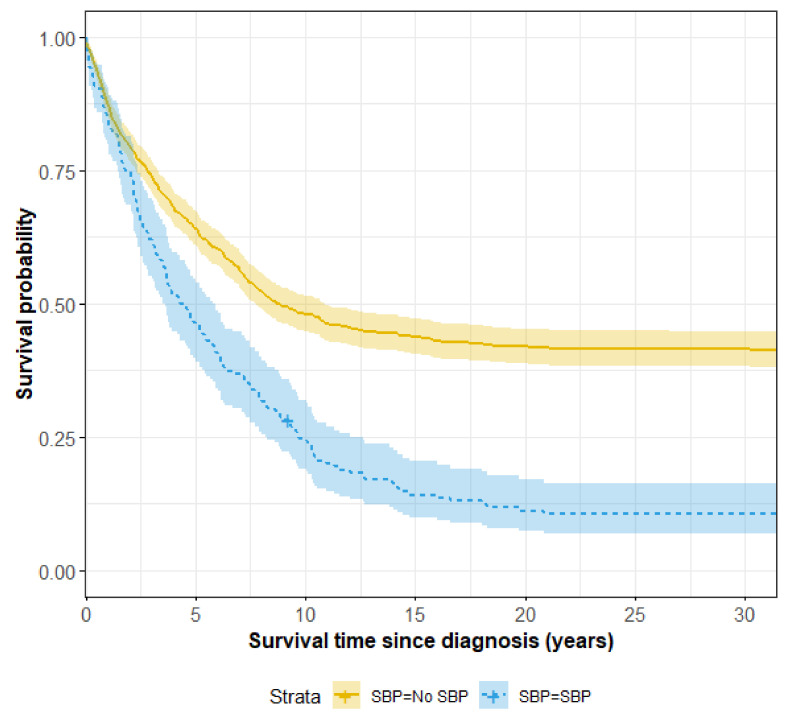
Long-term survival of cirrhotic patients with and without SBP.

**Table 1 jcm-11-00227-t001:** Demographic and clinical characteristics of the included patients.

Characteristics	SBP	without SBP	*p* Value
	*n* = 173 (%)	*n* = 869 (%)	
Sex—male (%)	(62) 108	(61) 533	0.7
Age—median (IQR)	60 (58, 76)	66 (55, 76)	0.7
Age at diagnosis cirrhosis median (IQR)	60 (51, 70)	59 (51, 72)	0.9
Ethnicity—Bedouin	16 (9.2)	78 (9)	0.9
Etiology of cirrhosis			
Hepatitis B	13 (7.5)	89 (10)	0.3
Hepatitis C	72 (42)	314 (36)	0.2
Fatty liver	32 (18)	151 (17)	0.5
Wilson disease	0 0	6 6 (0.7)	0.6
Alcoholic liver Disease	16 (9.2)	123 (14)	0.08
Autoimmune Hepatitis	5 (2.9)	19 (2.2)	0.6
PBC	1 (0.6)	15 (1.7)	0.2
PSC	2 (1.2)	2 (0.2)	0.1
Cryptogenic	17 (9.8)	83 (9.6)	>0.9
HCV and alcoholic	9 (5.2)	32 (3.7)	0.3
HCV and HBV	2 (1.2)	11 (1.3)	>0.9
HBV and alcoholic	1 (0.6)	11 (1.3)	0.7
Liver biopsy	40 (23)	221 (26)	0.5
Esophageal varices	120 (71)	452 (53)	<0.001
Esophageal varices bleeding	67 (41)	209 (26)	<0.001
Hepatocellular carcinoma (HCC)	22 (13)	128 (15)	0.5
Liver Transplantation	13 (7.6)	43 (5)	0.2
Death	157 (91)	515 (60)	<0.001
Age at death, Median (IQR)	66 (55, 76)	68 (59, 78)	0.2
Hb (gr/dL)	9.3 (8.3, 11)	11.2 (9.2, 3.2)	<0.001
WBC (10^3^/uL)	8 (5, 14)	7 (5, 10)	0.008
PLT (10^3^/uL)	80 (48, 130)	109 (72, 163)	<0.001
ALT (U/L)	30 (16, 58)	27 (17, 50)	0.5
AST (U/L)	64 (33, 109)	43 (28, 84)	0.001
Alkaline Phosphatase (U/L)	132 (85, 216)	114 (84, 178)	0.036
GGT (U/L)	56 (27, 121)	66 (34, 148)	0.015
Bili total (mg/dL), Last value	2.5 (1.5, 6.8)	1.4 (0.8, 3.1)	<0.001
Bili total (mg/dL), At diagnosis	1.3 (0.9, 2.3)	1.2 (0.7, 2)	0.043
Albumin (gr/dL), Last value	2.5 (2.0, 3.0)	2.9 (2.3, 3.8)	<0.001
Albumin (gr/dL), At diagnosis	3.2 (2.8, 3.6)	3.4 (2.9, 4.0)	<0.001
INR, Last value	1.7 (1.3, 2.4)	1.3 (1.1, 1.7)	<0.001
INR, At diagnosis	1.3 (1.1, 1.5)	1.2 (1.0, 1.4)	<0.001

Hb = hemoglobin, WBC = white blood cells, PLT = platelets, ALT = alanine aminotransferase, AST = aspartate aminotransferase, GGT = gamma-glutamyl transferase, Bili = bilirubin, and INR = international normalized ratio. All lab values are presented as median (IQR).

**Table 2 jcm-11-00227-t002:** Demographic and clinical characteristics of patients with positive and negative culture.

Characteristics	Negative Culture Neutrocytic Ascites *n* = 91 (%)	Positive Culture *n* = 82 (%)	*p*-Value
S–male	50 (55)	59 (72)	0.018
Age	68 (58, 77)	64 (54, 75)	0.066
Age at diagnosis	62 (53, 74)	58 (48, 67)	0.033
Ethnicity	7 (7.8)	8 (9.8)	0.8
Etiology of Cirrhosis			
Hepatitis B	9 (9.9%)	4 (4.9%)	0.2
Hepatitis C	44 (48.3%)	28 (34%)	0.05
Fatty Liver	14 (15%)	19 (23%)	0.2
Alcoholic Liver Cirrhosis	7 (7.8%)	9 (11%)	0.5
Cryptogenic	10 (11%)	6 (7.7%)	
Liver Biopsy	17 (18.7%)	23 (28%)	0.14
Esophageal Varices	58 (63.7%)	61 (74%)	0.3
Esophageal Varices Bleeding	27 (29.7%)	39 (49%)	0.034
HCC	12 (13%)	10 (12%)	0.8
Acute renal failure	35 (38.5)	46 (56)	0.023
Death	81 (89%)	74 (90%)	>0.9
HB	10.00 (8.8, 11.17)	9.00 (7.90, 10.47)	0.001
WBC	9 (5, 13)	7 (4, 14)	0.2
PLT	94 (55, 143)	66 (44, 117)	0.011
INR last Value	1.55 (1.30, 2.08)	1.7 (1.40, 2.58)	0.029
INR at diagnosis	1.30 (1.2, 1.50)	1.30 (1.10, 1.50)	0.4
Child-Pugh Score last value	10 (9, 12)	11 (9, 12)	0.020
Child-Pugh Stage last			0.091
A	10 (11%)	4 (4.9%)	
B	30 (33%)	20 (24%)	
C	52 (57%)	58 (71%)	

All lab values and Child-Pugh Score are presented as median (IQR).

**Table 3 jcm-11-00227-t003:** Bacterial resistances of germs against specific antibiotics.

Antibiotics Resistance	*E. coli n* = 31	*Klebsiella n* = 4	*Pseudomonas n* = 1	*Strep. pneum n* = 4	*Staph. areus* *n* = 7	*CONS*, *n* = 13	*Citrobacter n* = 2	*Acinetobacter* *n* = 3	*Entero**bacter* *n* = 2	*Entero**coccus* *n* = 6	*Strepviridans* *n* = 4
Ampicillin	22 (71%)	4 (100%)		0		2 (18%)	2 (100%)	2 (67%)	1 (100%)	3 (50)	0
Cefuroxime	16 (52%)	2 (50%)		0	1 (25%)	2 (18%)	1 (50%)	2 (100%)	2 (100%)		0
Rocephin	16 (52%)	2 (50%)		0	1 (20%)	2 (18%)	1 (50%)	2 (100%)	1 (50%)		0
Ceftazidime	16 (52%)	2 (50%)	0	0	1 (25%)	2 (18%)	0 (0%)	3 (100%)	2 (100%)		0
Gentamicin	7 (23%)	0	0	0	1 (14%)	2 (17%)	0 (0%)	3 (100%)	2 (100%)	2 (40)	0
Cotrimoxazole	15 (48%)	2 (50%)		0 (0%)	0	2 (17%)	0 (0%)	3 (100%)	1 (50%)		0
Amoxicillin/Clavulanic acid	13 (42%)	0		0	1 (25%)	2 (18%)	1 (50%)	2 (100%)	1 (50%)	0	0
Piperacillin	15 (54%)	1 (25%)	0	0	1 (25%)	2 (18%)	0 (0%)	3 (100%)	1 (50%)	0	0
Amikacin	0	0	0	0	1 (25%)	2 (18%)	0 (0%)	1 (50%)	0	0	0
Ciprofloxacin	15 (48%)	0	0				0 (0%)	3 (100%)	1 (50%)		
Meropenem	0	0	0	0	1 (25%)	2 (18%)	0 (0%)	3 (100%)	0		0
Piperacillin-Tazobactam	1 (3.2%)	0	0	0		2 (18%)	0 (0%)	3 (100%)	1 (50%)	0	0
Tobramycin	7 (25%)	0	0	0	1 (25%)	2 (18%)	0 (0%)	2 (100%)	0	0	0
Vancomycin				0	0	0				0	0

**Table 4 jcm-11-00227-t004:** COX regression model for mortality of patients with cirrhosis.

Characteristic	HR	95% CI	*p*-Value
SBP	1.62	(1.47, 1.78)	<0.001
Sex—Male	1.20	(1.11, 1.31)	0.03
Age	1.01	(1.00, 1.01)	<0.001
Ethnicity—Bedouin	0.67	(0.57, 0.8)	0.02
Anemia (Hemoglobin < 12)	2.71	(2.44, 3)	<0.001
Platelets	1.00	(1, 1)	0.67
Albumin < 3.5 at diagnosis	1.73	(0.57, 0.73)	<0.001
INR > 1 at diagnosis	1.40	(1.32, 1.63)	<0.001
Hepatitis B	0.68	(0.58, 0.8)	0.02
Hepatitis C	0.85	(0.76, 0.95)	0.14
Fatty liver	0.80	(0.7, 0.91)	0.09
Alcoholic liver disease	1.08	(0.94, 1.23)	0.59
Esophageal Varices	0.94	(0.87, 1.03)	0.50
HCC	1.54	(1.39, 1.72)	<0.001

HR = hazard ratio, and CI = confidence interval.

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
