# Peer review of "Spontaneous Bacterial Peritonitis among Cirrhotic Patients: Prevalence, Clinical Characteristics, and Outcomes"

_jcm, 2021, doi:10.3390/jcm11010227_

Round 1

Reviewer 1 Report

I reviewed the manuscript “Spontaneous Bacterial Peritonitis among cirrhotic patients: prevalence, clinical characteristics, and outcomes” in JCM

This topic is one of interesting issues in clinical practice. Authors clearly described the etiologies of SBP and analyzed mortality. I fully agree with suggestion in one of discussion points “awareness of local antimicrobial susceptibilities is key to the successful treatment of SBP infections”

It would be even better if authors add some points to this paper.

First, Please describe all-cause mortality according to liver related vs. non-liver related mortality among SBP compared with non-SBP patients. Authors suggested “explanation of high rate of mortality in our study in comparison the previous studied could be contributed to the increasing multidrug resistance strains.”

In addition, explain whether mortality may be associated with SBP as you suggest.

Seconds, in table 4 showed Hemoglobin also is significant risk factors for mortality(P<0.001). Why? This factor is not included in result and did not explain about it.

Reviewer 2 Report

Hi

This is an important area of research; however, it is not new and had been dealt with  before many times. It is accepted to re-do in case you expect new findings; which is not the case in this article.

The authors, although retrospectively extend their data since 1996 for 24ys; they do not get any benefit from this long period except large number of patients??!! It might be of great benefit if they subdivide this long period of time and compare their data before and after to get more valuable information e.g. before and after the use of quinolones as a prophylactic for SBP.

Statistically, they do not show us the sample size and how they calculate it.

In the result section, the tables are very bad editing manner???!!! In table 1 and 2 , they put numbers between brackets with no clear characteristics i.e is it mean+_ SD or median, range or %!!! all these had to be added in the first column (characteristics).

Table 3 needs new editing and remove all NA which carry no meaning. If they keep this NA, they have to write its meaning below the table.

Table 4 shows statistical significance of age, sex (needs < to be added), hemoglobin, albumin, INR (needs < to be added) and HCC. Strange enough, the authors mention only some of these items to be significant and did not mention the other factors!! needs explanation.

Spilling mistakes e.g line 149 better to be written cirrhotic patients. Line 153 alle??!! I think it is all.

References: 17 needs re-editing

                   18 is not present in the discussion at all!!!!

To conclude, there is no novelty at all. All the conclusions they came to are well known facts in this field. 

Round 2

Reviewer 1 Report

There is no additional revision point in this revised manuscript.